# TEACHING CONSENSUS RULES (AND EXCEPTIONS) TO LLMs FOR TRUSTWORTHY MEDICAL REASONING

## ABSTRACT

Machine learning for early prediction in medicine has recently shown breakthrough performance, however, the focus on improving prediction accuracy has led to a neglect of faithful explanations that are required to gain the trust of medical practitioners. The goal of this paper is to teach LLMs to follow medical consensus guidelines step-by-step in their reasoning and prediction process. Since consensus guidelines are ubiquitous in medicine, instantiations of verbalized medical inference rules to electronic health records provide data for fine-tuning LLMs to learn consensus rules and possible exceptions thereof for many medical areas. Consensus rules also enable an automatic evaluation of the model's inference process regarding its derivation correctness (evaluating correct and faithful deduction of a conclusion from given premises) and value correctness (comparing predicted values against real-world measurements). We exemplify our work using the complex Sepsis-3 consensus definition. Our experiments show that small fine-tuned models outperform one-shot learning of considerably larger LLMs that are prompted with the explicit definition and models that are trained on medical texts including consensus definitions. Since fine-tuning on verbalized rule instantiations of a specific medical area yields nearly perfect derivation correctness for rules (and exceptions) on unseen patient data in that area, the bottleneck for early prediction is not out-of-distribution generalization, but the orthogonal problem of generalization into the future by forecasting sparsely and irregularly sampled clinical variables. We show that the latter results can be improved by integrating the output representations of a time series forecasting model with the LLM in a multimodal setup.

## 1 INTRODUCTION

Medical consensus definitions are guidelines stated by a representative group of experts on how to diagnose and treat a disease based on clinical evidence. From the perspective of logical inference, consensus guidelines include deductive and inductive inference rules. Deductive rules have the form of if-then relations where the conclusion is certain given the correctness of the premise, for example, in mapping thresholds on clinical measurements to step functions of diseases. If used for early prediction purposes (a.k.a. prognosis), inductive inference rules are required where the if-then relation between premise and conclusion is probabilistic. An illustrative example for a complex consensus guideline is the Sepsis-3 definition that identifies an organ dysfunction as an acute change in total SOFA score $\geq 2$ points consequent to an infection (Singer et al., 2016; Seymour et al., 2016). The SOFA (Sepsis-Related Organ Failure Assessment) score itself constitutes a consensus definition that is based on definitions for six organ systems, each defining thresholds on particular clinical variables observed during a 24h window ((Vincent et al., 1996), see Table 5 in Appendix A.1). These calculations comprise a logical rule system where time series forecasting (TSF) of clinical variables represents inductive rules, which are composed with deductive rules that calculate extrema over time, map clinical measurements onto step functions, and calculate changes over time (see Figure 1).

The goal of this work is to teach LLMs to follow the deductive and inductive rules of medical consensus guidelines step-by-step in their prediction and reasoning process[1], in order to foster the trust of medical practitioners in the generated diagnosis or prognosis. We exemplify our work

---

[1]In the following, in order to avoid an anthropomorphic attribution of "reasoning" capabilities to LLMs, we will speak of medical "rules" that are taught, and of an "inference process" that is being evaluated for the LLM.

using the complex Sepsis-3 consensus definition. Instantiations of verbalized medical inference rules to clinical patient data allow the model to learn the consensus rule and possible exceptions to the rule from patient examples in many medical areas (see Figure 2). Our work transfers learning of compositional inference tasks from mathematical problems to medical inference, with several advantages.

First, consensus definitions are ubiquitous in medicine, ranging from guidelines for mental disorders (American Psychiatric Association, 2013) to neurological (McDonald et al., 2001) and physiological diseases (KDIGO Acute Kidney Injury Work Group, 2012). For each area, verbalization of consensus rules can be done automatically by using templates (which can themselves be generated automatically by using LLMs) that describe each step of an application of a consensus rule system to patient data. Furthermore, human curation can be integrated by annotating verbalized inference chains with possible exceptions and corrections to the rule. In contrast to prompting LLMs, supervised fine-tuning allows learning of rules and exceptions from example instantiations to patient data.

Second, the use of consensus rules enables an exact and automatic evaluation of the model's inference process against a trusted medical gold standard. We present an evaluation setup that differentiates the derivation correctness of a trained model — measuring a model's ability to learn to correctly deduce a conclusion from a given premise — from a model's value correctness — comparing the numerical value predicted in each inference step against real-world measurements. An evaluation of derivation correctness also can be seen to measure the faithfulness of the model's inference process in the sense of checking for an accurate representation of the reasoning process behind the model's prediction.

Our results show that small fine-tuned models (LLaMA 8B parameters) outperform one-shot learning of considerably larger LLMs (LLaMA 70B parameters) that are given the explicit definition in the prompt, and LLMs that are trained on medical texts including the original consensus definitions (Me-LLaMA 8B parameters) under all evaluation metrics. In particular, fine-tuned LLMs show nearly perfect generalization to unseen patient data with respect to derivation correctness of rules and exceptions. This demonstrates that an adaptation of an LLM to verbalized rule instantiations for a specific medical area likely guarantees consistent results in that area, whereas training or prompting models with the abstract definition texts is not sufficient. We conjecture that generalization across consensus definitions in areas as different as psychiatry (based on interviews (American Psychiatric Association, 2013)), neurology (based partially on magnetic resonance imaging (McDonald et al., 2001)), or physiological diseases (based on vital signals and lab tests (KDIGO Acute Kidney Injury Work Group, 2012; Cederholm et al., 2019; Singer et al., 2016; Seymour et al., 2016)) is difficult to bridge without supervision. The bottleneck is thus not out-of-distribution generalization (Chu et al., 2025), but the orthogonal problem of generalization into the future, i.e., the inherent complexity of time series forecasting of sparsely and irregularly sampled clinical variables. The latter results can be improved by moving from a text-based encoding of clinical measurements to a multi-modal approach where the output representations of a TSF model are fed as additional input to the LLM.

## 2 RELATED WORK

**Reasoning in LLMs** Recent evaluations of the "reasoning" process of LLMs have shown that they are able to correctly solve sub-tasks of multi-step inference tasks, but fail to compose them into a fully correct inference path, especially in extrapolation to more complex out-of-distribution data (Dziri et al., 2023; Zhang et al., 2023; Saparov & He, 2023; Yang et al., 2024; Mondorf & Plank, 2024). For compositional inference problems in mathematics or logics, fine-tuning on so-called "scratchpads" has been shown to yield consistently strong generalization (Nye et al., 2021; Hochlehnert et al., 2025). To our knowledge, our work is the first one to apply an automatic generation of scratchpads for fine-tuning and step-by-step evaluation to the inference process of medical LLMs. Recent works have addressed the aspect of faithfulness of chain-of-thought explanations of LLMs, asking whether they accurately represent the underlying inference process of the model (Jacovi & Goldberg, 2020). Faithfulness in chain-of-thought reasoning has been enforced by incorporating deterministic solvers into the inference process (Lyu et al., 2023; Xu et al., 2024). Evaluation of faithfulness has been done manually for LLMs with explicit (Turpin et al., 2023) and implicit biases (Arcuschin et al., 2025), or by defining automatic evaluation metrics of chain-of-thought inference processes (Lanham et al., 2023; Wang et al., 2025; Chen et al., 2025). The latter metrics are based on accuracy or confidence of

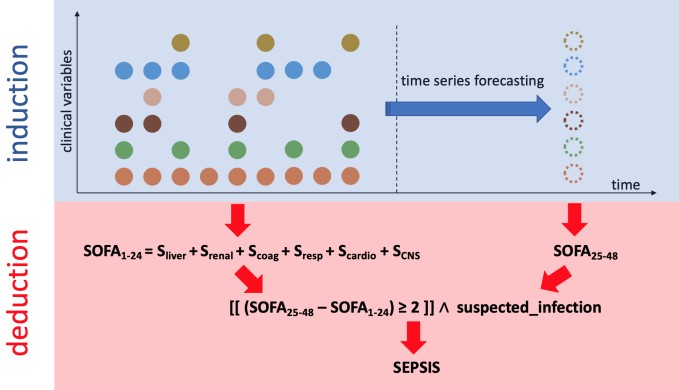

Figure 1: Inductive and deductive inference rules in the Sepsis-3 Consensus Definition (Singer et al., 2016; Seymour et al., 2016). Deductive rules calculate extrema over time, map thresholds onto step functions for SOFA scores, and calculate total SOFA and changes over time. Inductive rules involve time series forecasting of clinical variables 24 hours into the future.

the generated LLM outputs, and can thus be considered "reference-free" evaluation metrics, whereas a consensus rule can be used as a unique gold standard reference for evaluation.

**Medical Reasoning**   Medical reasoning has been characterized to include two contrasting mechanisms, termed analytic versus non-analytic strategies (Eva, 2005), causal versus example-based knowledge (Norman, 2005), or contrasting external clinical evidence with individual clinical expertise (Sackett et al., 1996). To our knowledge, no attempt has been made so far to differentiate these aspects in teaching and evaluating LLMs for healthcare. Instead, most evaluations are based on medical question-answering (Liu et al., 2023; Singhal et al., 2025), while it has been shown that the vast majority of questions in medical QA benchmarks can be answered by factual recall, not requiring multi-step inference at all (Thapa et al., 2025). While excelling in medical question-answering tasks, few-shot LLMs suffer from prompt brittleness (Hochlehnert et al., 2025; Reese et al., 2024) and fail in basic TSF tasks (Merrill et al., 2024; Tan et al., 2024) and lack adherence to consensus guidelines (Hager et al., 2024). This discrepancy calls for evaluations of medical LLMs that go beyond the final result and instead evaluate the inference process itself. Various approaches have been presented to teach LLMs medical inference rules, e.g., by pre-training or fine-tuning on medical texts including consensus guidelines (Chen et al., 2023; Xie et al., 2025), or by incorporating rule-based knowledge in knowledge graphs (Wu et al., 2025), reward models (Yun et al., 2025; Chen et al., 2024; Zhang et al., 2025), or by manual annotation by clinical experts (Liu et al., 2025). However, with the exception of the last work which relies on manual evaluation, all works are evaluated on standard medical question-answering benchmarks. Our work uses trusted consensus rules as training data and, instantiated to different patients, as gold standard references for evaluation, and obviates the need for manual evaluation.

**Early Prediction of Sepsis**   Machine learning for early prediction of sepsis starts by labeling clinical data[2], by applying a Sepsis consensus definition [3] to future clinical observations that are unseen to the model, and then training models to predict the label based on clinical measurements observed several hours before. Examples are the machine learning approaches taken by the 104 participants in the PhysioNet Challenge on "Early Prediction of Sepsis from Clinical Data" (Reyna et al., 2019), or the 21 approaches described in a recent overview (Moor et al., 2021). Machine learning for early prediction of sepsis includes a broad variety of learning algorithms, from linear learners to deep learning (see Reyna et al. (2019) and Moor et al. (2021)), up to most recent works using Transformers (Choi et al., 2024), graph neural networks (Yin et al., 2025), and pretrained LLMs (Li et al., 2024). The setup of predicting a label that is the outcome of a consensus definition can be abstractly viewed as the prediction of an effect that is caused by applying a consensus definition to future values of

---

[2]E.g., MIMIC-III (Johnson et al., 2016), MIMIC-IV (Johnson et al., 2023), or eICU (Pollard et al., 2018).
[3]Sepsis-3 or Sepsis-2 (Levy et al., 2003; Dellinger et al., 2013) in earlier works.

clinical variables. Another option is to directly predict the causes by forecasting clinical variables, and determine the effect by applying the consensus definition to the forecasted values (Staniek et al., 2024). Our approach is most similar to the latter, with the important difference that our approach not only learns rules, but also exceptions thereof from patient data.

## 3 TEACHING LLMs TO DIAGNOSE ACCORDING TO THE SEPSIS-3 DEFINITION

**Deductive Inference**  Sepsis-3 contains several deductive inference steps: A first step is a calculation of "worst" values (minima or maxima) of clinical variables over 24 hours. The crucial deductive task in Sepsis-3 can be seen as a deductive syllogism consisting of rules (the major premises) that are instantiated to thresholds on clinical variables (the minor premises) that are mapped to step functions for six SOFA subscores, yielding scores from $0 - 4$ (the conclusions). Each SOFA subscore corresponds to an organ system, namely the central nervous system, and the cardiovascular, respiratory, coagulation, liver, and renal organ systems. The rules themselves are abbreviated as $S_{CNS}$, $S_{cardio}$, $S_{resp}$, $S_{coag}$, $S_{liver}$, and $S_{renal}$ in the lower part of Figure 1 and shown in the columns in Table 5 in Appendix A.1. The total SOFA score is calculated by summing these six subscores to a score ranging from $0 - 24$, once for clinical measurements during the first 24 hours ($SOFA_{1-24}$), and for predicted measurements in the next 24 hours ($SOFA_{25-48}$). A further deductive inference step is the computation of a change $\geq 2$ as

$$SOFA_{diff} := [\![(SOFA_{25:48} - SOFA_{1:24}) \geq 2]\!], \tag{1}$$

where $[\![a]\!] = 1$ if $a$ is true, 0 otherwise. Finally, a binary Sepsis label is assessed by combining the indicator function in Equation 1 with a binary indicator suspected_infection[4], yielding

$$SEPSIS := SOFA_{diff} \land suspected\_infection. \tag{2}$$

**Inductive Inference**  Sepsis-3 is laid out to detect a life-threatening organ dysfunction by an acute change in the total SOFA score. Measuring an increase in SOFA score over time requires TSF of future values of a set of clinical features $F$ (see upper part of Figure 1), and an application of the SOFA definition to current and future clinical values. Formally, given a representation $x$ of an input time series, an output vector of predicted clinical values $\hat{y}_t \in \mathbb{R}^{|F|}$ is produced. The features used in our experiments are 131 clinical measurements extracted from the MIMIC-III database (Johnson et al., 2016). A full list is given in Appendix A.4. The TSF task in our experiments is defined to predict the next 24 hours from a history of preceding 24 hours. Dedicated models to perform the TSF task are described in Appendix A.2.

**Teaching Verbalized Consensus Rules to Autoregressive LLMs**  An approach to teaching LLMs to generate a chain of inference rules that adheres to a certain definition is to deploy an autoregressive architecture where next-token prediction is based on a history of previously predicted tokens, and token-wise errors on a target inference chain can be backpropagated through the system. This procedure coincides with standard supervised fine-tuning, where an input question, a gold-standard verbalization of the inference process, and the answer, is provided to the model during training. An example for fine-tuning data including inference rules following the Sepsis-3 definition is shown in Figure 2, left column. The data starts with a verbalization of a sparse multivariate input time series of measurements for a 24 hour window, followed by an instruction to classify the patient as septic in the next 24 hours, given information about suspected infection[5]. The third text block in the fine-tuning data consists of a gold standard inference chain explaining the scores of the six SOFA systems from the corresponding clinical measurements, and computing their sum, given the measurements of the first 24 hour window. The fourth text block includes forecasts of the clinical variables relevant for SOFA for the next 24 hours, and the corresponding inference about the six SOFA subscores, and their sum. The last text block contains a gold standard inference about the Sepsis prediction and the responsible SOFA system, if applicable.

---

[4]Following Singer et al. (2016); Seymour et al. (2016), a suspected infection is defined as a combination of antibiotics treatment and blood cultures, starting within the first 24 hours after admission.

[5]The onset of suspicion of infection is thus given prior to the onset of salient organ dysfunction in our approach, corresponding to labeling scheme H1 of Cohen et al. (2024), and consistent with other work on early prediction of sepsis where suspected infection is included in the list of input features (Nemati et al., 2018).

Patient is 75.0 years old and is male. Given all the information in this text, answer the question at the end. Here are the measurements: DBP at time -22.37: 49.0, SBP at time -22.37: 105.0, DBP at time -20.37: 52.0, GCS_eye at time -20.37: 4.0, GCS_motor at time -20.37: 6.0, GCS_verbal at time -20.37: 1.0, SBP at time -20.37: 117.0, DBP at time -19.37: 56.0, FiO2 at time -19.37: 0.5, SBP at time -19.37: 127.0, DBP at time -18.37: 43.0,...

Now answer the following question: The doctors suspect an infection, based on this information and the other information in this text, will the patient be classified as septic tomorrow?

First we need to calculate the SOFA scores given the extracted values. The SOFA scores for the current time are the following: The minimum value of GCS_eye is 4.0, GCS_motor is 6.0 and GCS_verbal is 1.0, this produces the sum 11.0 and means the CNS SOFA is 2. Because minimum MAP is 55.333, max Dopamine is 0, max Dobutamine is 0, max Epinephrine is 0 and max Norepinephrine is 0 with a patient weight of 62.8 kg, the cardiovascular SOFA is 1. Given that minimum PO2 is 100.0 and minimum FiO2 is 0.5 the calculated PAO2FIO2 is 200.0, this means the respiratory SOFA is 2. Because the minimum Platelet count is 310.0 the coagulation SOFA is 0. The maximum Bilirubin (Total) is 1 leading to a liver SOFA of 0. Because total Urine output is 1095.0 and maximum creatinine in the blood is 0.4 the renal SOFA is 0. To summarize: the patient has a total SOFA score of 5.

Now we need to calculate the SOFA scores with forecasted values. The SOFA scores in the future based on the forecasted values are the following: The minimum value of GCS_eye will be 4.0, GCS_motor will be 6.0 and GCS_verbal will be 1.0, this produces the sum 11.0 and means the CNS SOFA will be 2. Because future minimum MAP will be 65.333, future max Dopamine will be 0, future max Dobutamine will be 0, future max Epinephrine will be 0 and future max Norepinephrine will be 0 with a patient weight of 62.8 kg, the cardiovascular SOFA will be 1. Given that minimum PO2 will be 100.0 and minimum FiO2 will be 0.5 the forecasted PAO2FIO2 will be 200.0, this means the respiratory SOFA will be 2. Because the Platelet count will be 310.0 the coagulation SOFA is going to be 0. The maximum Bilirubin (Total) will be 1 leading to a liver SOFA of 0. Because Urine output will be 150.0 and maximum creatinine in the blood will be 0.4 the renal SOFA will be 4. **To summarize: the patient will have a future total SOFA score of 9.**

**This calculation means that the patient will likely experience a kidney failure since SOFA increased by 4. The patient will develop sepsis in the next 24 hours, because total SOFA increased by 4 and infection is suspected.**

Patient is 75.0 years old and is male. Given all the information in this text, answer the question at the end. Here are the measurements: DBP at time -22.37: 49.0, SBP at time -22.37: 105.0, DBP at time -20.37: 52.0, GCS_eye at time -20.37: 4.0, GCS_motor at time -20.37: 6.0, GCS_verbal at time -20.37: 1.0, SBP at time -20.37: 117.0, DBP at time -19.37: 56.0, FiO2 at time -19.37: 0.5, SBP at time -19.37: 127.0, DBP at time -18.37: 43.0,...

Now answer the following question: **The patient has an existing precondition given by the ICD-10 code N18.9.** The doctors suspect an infection, based on this information and the other information in this text, will the patient be classified as septic tomorrow?

First we need to calculate the SOFA scores given the extracted values. The SOFA scores for the current time are the following: The minimum value of GCS_eye is 4.0, GCS_motor is 6.0 and GCS_verbal is 1.0, this produces the sum 11.0 and means the CNS SOFA is 2. Because minimum MAP is 55.333, max Dopamine is 0, max Dobutamine is 0, max Epinephrine is 0 and max Norepinephrine is 0 with a patient weight of 62.8 kg, the cardiovascular SOFA is 1. Given that minimum PO2 is 100.0 and minimum FiO2 is 0.5 the calculated PAO2FIO2 is 200.0, this means the respiratory SOFA is 2. Because the minimum Platelet count is 310.0 the coagulation SOFA is 0. The maximum Bilirubin (Total) is 1 leading to a liver SOFA of 0. Because total Urine output is 1095.0 and maximum creatinine in the blood is 0.4 the renal SOFA is 0. To summarize: the patient has a total SOFA score of 5.

Now we need to calculate the SOFA scores with forecasted values. The SOFA scores in the future based on the forecasted values are the following: The minimum value of GCS_eye will be 4.0, GCS_motor will be 6.0 and GCS_verbal will be 1.0, this produces the sum 11.0 and means the CNS SOFA will be 2. Because future minimum MAP will be 65.333, future max Dopamine will be 0, future max Dobutamine will be 0, future max Epinephrine will be 0 and future max Norepinephrine will be 0 with a patient weight of 62.8 kg, the cardiovascular SOFA will be 1. Given that minimum PO2 will be 100.0 and minimum FiO2 will be 0.5 the forecasted PAO2FIO2 will be 200.0, this means the respiratory SOFA will be 2. Because the Platelet count will be 310.0 the coagulation SOFA is going to be 0. The maximum Bilirubin (Total) will be 1 leading to a liver SOFA of 0. Because Urine output will be 150.0 and maximum creatinine in the blood will be 0.4 the renal SOFA will be 4. **To summarize: the patient will have a future total SOFA score of 5**.

**The patient will not develop sepsis in the next 24 hours, because total SOFA increased by 0 and infection is suspected.**

Figure 2: Fine-tuning data including a verbalization of Sepsis-3 inference (left column) and inference under an exception due to medical preconditions (right column). Differences are shown in bold blue font. The general prompt is shown above the horizontal line, the gold standard answer below.

**Handling Exceptions to the Rules** Since consensus rules cannot capture every individual disease progression, it is important to consider the possibility of manual corrections to consensus-based inference by adding exceptions to the inference rules. A possible example are medical preconditions with specific treatments, for example, dialysis in case of chronic kidney disease. While there is clear evidence that incorporating pre-existing medical conditions (comorbidities / chronic organ failure) can improve sepsis prediction (Sarraf et al., 2024; Christensen et al., 2023), there is no consensus on how certain preconditions which will alter clinical measurements and thus the SOFA score. In case of a chronic kidney disease, a conservative exception might be to decide to disregard the kidney SOFA in the calculation of total SOFA, leading to an exception to the Sepsis-3 definition. We test this hypothetical scenario by synthetic data. We synthesized fine-tuning data including exceptions to the rule by adding preconditions for five organ systems in form of ICD-10 codes. During data generation, each datapoint was randomly assigned a precondition, either one of the predefined types or the no-precondition option, with equal probability. An example for a verbalization is shown in Figure 2, right column. The format is the same as for inference according to the definition, except for the inclusion of a precondition in the second text block — in our example, the ICD code N18.9 indicating chronic kidney disease — implying an exception to the rule to disregard the SOFA score of the corresponding organ system in the fourth and fifth text blocks. We prepared the data such

Table 1: Derivation correctness of predicted values. Forced derivation correctness in brackets.

| | variable | one-shot | one-shot-70B | me-llama | deepseek | fine-tuned | pipeline | multimodal |
|---|---|---|---|---|---|---|---|---|
| current | $S_{CNS}$ | 0.693 | 0.803 | 0.520 | 0.288 | 1.000 | 1.000 | 1.000 |
| | $S_{cardio}$ | 0.516 | 0.696 | 0.258 | 0.338 | 0.998 | 0.998 | 0.975 |
| | $S_{resp}$ | 0.561 | 0.302 | 0.404 | 0.279 | 0.995 | 0.996 | 0.996 |
| | $S_{coag}$ | 0.599 | 0.763 | 0.511 | 0.381 | 0.966 | 0.966 | 0.970 |
| | $S_{liver}$ | 0.537 | 0.864 | 0.601 | 0.336 | 0.993 | 0.992 | 0.992 |
| | $S_{renal}$ | 0.537 | 0.730 | 0.396 | 0.256 | 1.000 | 0.998 | 1.000 |
| | $SOFA_{1:24}$ | 0.543 | 0.893 | 0.777 | 0.165 | 1.000 | 1.000 | 0.999 |
| future | $S_{CNS}$ | 0.504 (0.827) | 0.545 (0.895) | 0.399 (0.786) | 0.261 (0.823) | 1.000 (1.000) | 1.000 (1.000) | 1.000 (1.000) |
| | $S_{cardio}$ | 0.580 (0.694) | 0.679 (0.714) | 0.353 (0.647) | 0.269 (0.671) | 0.987 (0.997) | 0.975 (0.997) | 0.965 (0.997) |
| | $S_{resp}$ | 0.530 (0.756) | 0.307 (0.827) | 0.384 (0.733) | 0.266 (0.722) | 0.995 (0.996) | 0.996 (0.996) | 0.995 (0.996) |
| | $S_{coag}$ | 0.661 (0.786) | 0.754 (0.818) | 0.519 (0.801) | 0.306 (0.722) | 0.967 (1.000) | 0.970 (0.999) | 0.966 (1.000) |
| | $S_{liver}$ | 0.534 (0.835) | 0.847 (0.969) | 0.599 (0.657) | 0.335 (0.776) | 0.992 (0.999) | 0.992 (0.999) | 0.992 (1.000) |
| | $S_{renal}$ | 0.554 (0.708) | 0.737 (0.773) | 0.414 (0.690) | 0.251 (0.634) | 1.000 (1.000) | 1.000 (1.000) | 1.000 (1.000) |
| | $SOFA_{25:48}$ | 0.518 (0.875) | 0.868 (0.984) | 0.768 (0.992) | 0.148 (0.550) | 1.000 (1.000) | 0.999 (1.000) | 1.000 (1.000) |
| | $SOFA_{diff}$ | 0.845 (0.740) | 0.891 (0.477) | 0.769 (0.414) | 0.503 (0.351) | 1.000 (1.000) | 1.000 (1.000) | 1.000 (1.000) |
| | SEPSIS | 0.859 (0.682) | 0.909 (0.727) | 0.769 (0.688) | 0.738 (0.521) | 1.000 (1.000) | 1.000 (1.000) | 1.000 (1.000) |

that some preconditions are seen during training and testing, while we also chose some codes from the same ICD class as out-of-distribution data that were only seen at test time. The codes and their distribution are shown in Tables 9 and 10 in Appendix A.6.

## 4 EXPERIMENTAL SETUP

**Data and Models**  In our experiments, we use electronic health records (EHRs) from the MIMIC-III data (Johnson et al., 2016). After filtering for patients with an ICU stay of at least 24 hours with reported gender and age of at least 18 years, our dataset contained 44,858 ICU stays with 56 million data points. We split the data into partitions for fine-tuning (28,708), development (7,270), and testing (8,880). For computational reasons, we further subsampled 15,000 datapoints for fine-tuning, and 3,000 datapoints for development and testing, respectively. The resulting percentage of positive Sepsis cases in the test set was 7.33%. As features, we considered 131 clinical variables and the demographic variables gender and age (see Appendix A.4). This selection comprises all vital signs and laboratory values used in the PhysioNet challenge for early prediction of sepsis (Reyna et al., 2019), together with information on suspected infection as in Nemati et al. (2018). The time series of clinical observations in the data were split into a 24 hour observation window, followed by a 24 hour prediction window. During training and testing, we use a sliding window of 24 hours so that full admission days are given as input and output. Using the consensus definition given by Table 5 and Equation 2, SOFA scores and Sepsis label were calculated deterministically for the given data. Following this, features and SOFA scores were verbalized into texts as seen in Figure 2. In the calculation of the ground truth, missing values are being carried forward from the previous day only.

The basic LLM used in our experiments is a pretrained Llama-3 model with 8B parameters (Grattafiori et al., 2024) [6] used in **one-shot** mode. All one-shot experiments use the prompt shown in Appendix A.5 that includes the explicit Sepsis-3 rules together with an example instantiation to patient data. Our **fine-tuned** model uses LORA adapters (Hu et al., 2022) on verbalizations of inference processes. Furthermore, we use a dedicated medical TSF forecaster, pre-trained on MIMIC-III to improve the inductive inference part of the LLM (see Appendix A.2). The TSF model is used in two ways: First, we extract predictions from the forecaster and augment the prompt with those predictions in a **pipeline** approach. In this approach, the Llama-3 model is still finetuned using LORA, but the forecaster is kept fixed. Second, we use a **multimodal** setup to connect the forecaster with the Llama-3 model. We extract the full decoder prediction from the forecaster. The output vectors are then fed into a connector MLP to produce vectors with Llama-3 embedding dimensionality. These time series "token embeddings" are then prepended to the actual text embeddings. The LORA adapters in the Llama-3 model, the connector MLP, and the TSF model get updated during the finetuning process. A list of hyperparameter settings chosen on the validation set (except the default setting for the Llama-3 8B model) is given in Appendix A.3. For further comparison, we use an 8B parameters LLM pretrained on medical texts, including publications and wikipedia texts on consensus guidelines (**me-llama**

---

[6] https://huggingface.co/meta-llama/Llama-3.1-8B-Instruct

Table 2: Derivation correctness for in- and out-of-distribution exceptions to inference rules.

| variable | % changes ID | % changes OOD | ID score | OOD score |
|---|---|---|---|---|
| $SOFA_{1:24}$ | 0.404 | 0.424 | 1.000 | 1.000 |
| $SOFA_{25:48}$ | 0.433 | 0.452 | 1.000 | 1.000 |

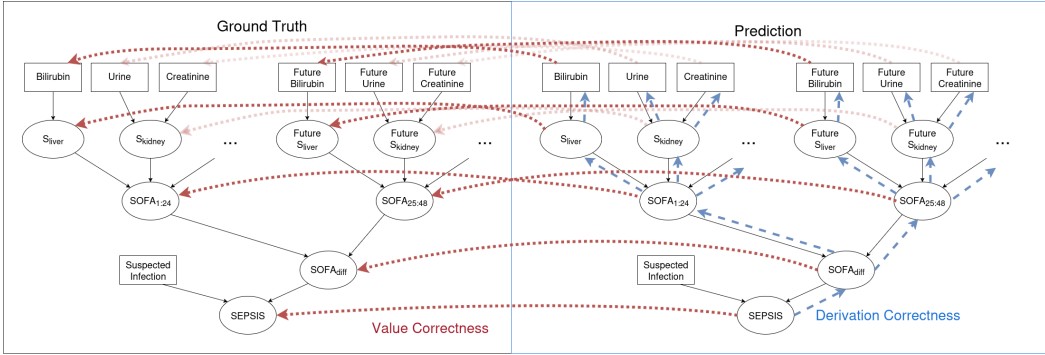

Figure 3: **Derivation correctness** (dashed blue arrows) checks for the predicted inference graph whether each child node (conclusion) follows from the parent node (premise) according to the consensus rule. **Value correctness** (dotted red arrows) maps each node in the predicted inference graph to its corresponding node in the ground truth graph, consisting of real-world clinical measurements for first and second 24 hours, and deterministic calculation of SOFA and SEPSIS on these values.

(Xie et al., 2025)[7]), and a reasoning-style LLM (**deepseek** DeepSeek-AI et al. (2025)[8], and 70B pretrained Llama-3 model (**one-shot-70B**[9]), all in one-shot mode.

**Evaluation Metrics** We evaluate the generated output of LLMs in three different ways. First, we evaluate the LLM output according to **derivation correctness** (dashed blue arrows in Figure 3) which checks whether each child node (conclusion) in the predicted inference graph follows from the parent node (premise) according to the consensus rule. This evaluation measures the ability of the LLM to learn the deductive inference rules of the consensus definition. Furthermore, it measures faithfulness of the model's inference process in the sense of checking for an accurate representation of the reasoning process behind the model's prediction. In order to avoid trivial cases of derivation correctness, for example, inference of unchanged SOFA scores from unchanged future measurement predictions, we measure in addition **forced derivation correctness**. This is done by forcing the decoder to use the correct inference chain up to a last measurement token of a SOFA system as context, then continuing the decoding process until the specified target node. Furthermore, we conduct an evaluation according to **value correctness** (dotted red arrows in Figure 3) which compares the numerical value of each partial inference node in the prediction graph with a corresponding real-world output. Since errors earlier in the inference necessarily propagate through the inference chain, this evaluation measures the utility of the LLM output in practical applications. Comparison in all three cases is done by parsing the LLM output for relevant keywords, extracting numbers, and checking if the predicted number lies in a specific interval around the true value. A match is considered positive if the result deviates from the true value by $5\%$ (see Appendix A.7 for a discussion). In case of value correctness, we calculate a contingency table by comparing the application of the Sepsis-3 definition to a model involving forecasted clinical measurements with Sepsis-3 calculated for real-world clinical measurements in the second 24 hours block. From this, metrics such as accuracy, specificity, sensitivity and F1 are computed.

---

[7] https://huggingface.co/YBXL/Med-LLaMA3-8B
[8] https://huggingface.co/deepseek-ai/DeepSeek-R1-Distill-Llama-8B
[9] https://huggingface.co/meta-llama/Llama-3.1-70B-Instruct

Table 3: Value correctness of predicted clinical variables compared to real-world measurements. All fine-tuning improvements over one-shot learning are statistically significant according to an approximate randomization test.

| | variable | one-shot | one-shot-70B | me-llama | deepseek | fine-tuned | pipeline | multimodal |
|---|---|---|---|---|---|---|---|---|
| current | GCS-eye | 0.855 | 0.982 | 0.764 | 0.608 | 0.998 | 0.997 | 0.998 |
| | GCS-motor | 0.914 | 0.980 | 0.733 | 0.616 | 0.998 | 0.998 | 0.998 |
| | GCS-verbal | 0.947 | 0.985 | 0.750 | 0.635 | 0.999 | 0.998 | 0.999 |
| | MAP | 0.036 | 0.219 | 0.046 | 0.089 | 0.993 | 0.976 | 0.994 |
| | Dopamine | 0.875 | 0.952 | 0.754 | 0.078 | 0.972 | 0.964 | 0.968 |
| | Dobutamine | 0.905 | 0.978 | 0.785 | 0.083 | 0.993 | 0.991 | 0.992 |
| | Epinephrine | 0.891 | 0.965 | 0.773 | 0.082 | 0.979 | 0.980 | 0.980 |
| | Norepinephrine | 0.809 | 0.875 | 0.692 | 0.076 | 0.909 | 0.895 | 0.906 |
| | Weight | 0.851 | 0.765 | 0.722 | 0.280 | 0.999 | 0.999 | 0.999 |
| | PaO2/FiO2 | 0.140 | 0.559 | 0.090 | 0.059 | 0.998 | 0.997 | 0.997 |
| | PaO2 | 0.533 | 0.563 | 0.352 | 0.288 | 0.998 | 0.997 | 0.997 |
| | FiO2 | 0.881 | 0.711 | 0.721 | 0.479 | 1.000 | 1.000 | 1.000 |
| | Platelet | 0.922 | 0.966 | 0.676 | 0.664 | 0.997 | 0.996 | 0.996 |
| | Bilirubin | 0.299 | 0.275 | 0.262 | 0.177 | 0.999 | 0.998 | 0.999 |
| | Urine | 0.102 | 0.262 | 0.095 | 0.037 | 0.966 | 0.761 | 0.880 |
| | Creatinine | 0.887 | 0.970 | 0.666 | 0.408 | 0.998 | 0.998 | 0.998 |
| future | GCS-eye | 0.679 | 0.710 | 0.525 | 0.357 | 0.712 | 0.743 | 0.789 |
| | GCS-motor | 0.783 | 0.805 | 0.620 | 0.398 | 0.811 | 0.832 | 0.869 |
| | GCS-verbal | 0.806 | 0.840 | 0.648 | 0.434 | 0.848 | 0.871 | 0.895 |
| | MAP | 0.110 | 0.197 | 0.125 | 0.064 | 0.274 | 0.289 | 0.294 |
| | Dopamine | 0.917 | 0.966 | 0.766 | 0.386 | 0.978 | 0.987 | 0.988 |
| | Dobutamine | 0.937 | 0.982 | 0.787 | 0.388 | 0.994 | 0.993 | 0.993 |
| | Epinephrine | 0.933 | 0.972 | 0.779 | 0.381 | 0.989 | 0.989 | 0.989 |
| | Norepinephrine | 0.861 | 0.887 | 0.710 | 0.350 | 0.913 | 0.914 | 0.913 |
| | Weight | 0.851 | 0.765 | 0.722 | 0.280 | 0.915 | 0.916 | 0.914 |
| | PaO2/FiO2 | 0.073 | 0.156 | 0.053 | 0.038 | 0.565 | 0.584 | 0.596 |
| | PaO2 | 0.181 | 0.179 | 0.148 | 0.098 | 0.616 | 0.659 | 0.715 |
| | FiO2 | 0.738 | 0.538 | 0.613 | 0.379 | 0.821 | 0.840 | 0.867 |
| | Platelet | 0.283 | 0.300 | 0.236 | 0.166 | 0.364 | 0.383 | 0.396 |
| | Bilirubin | 0.180 | 0.166 | 0.169 | 0.115 | 0.811 | 0.832 | 0.866 |
| | Urine | 0.064 | 0.077 | 0.069 | 0.023 | 0.133 | 0.152 | 0.183 |
| | Creatinine | 0.360 | 0.347 | 0.295 | 0.161 | 0.395 | 0.424 | 0.455 |

## 5 RESULTS AND DISCUSSION

**LLMs Shine in Deductive Inference According to Consensus Rules**  Our first research question asks whether LLMs can learn the deductive inference rules underlying medical consensus definitions by fine-tuning on verbalized instantiations of the inference process to patient data. Table 1 shows that all fine-tuned models, from basic fine-tuning with text encoding to pipeline and multimodal approaches, achieve nearly perfect derivation correctness. This includes mapping clinical variables to SOFA step functions for the first 24 hours (rows 1-7), mapping forecasted clinical variables to future SOFA scores (rows 8-15), and computing a Sepsis label (row 16). In contrast, one-shot learning does not perform nearly as good, even for models that include abstract consensus definitions in their training data (me-llama) or prompt and are significantly larger (one-shot-70B). Forced derivation correctness (in brackets in Table 1) shows that predictions using correct histories are very similar to using predicted histories. Again, all one-shot learners underperform compared to fine-tuning.

**LLMs Can Learn Exceptions to Consensus Rules**  The next question we ask is whether LLMs can learn exceptions to the inference rules that might conflict with the consensus definition. In our hypothetical scenario, we synthesize examples where ICD codes indicate a medical precondition related to an organ system, with the exception to the rule implying that the SOFA score for the respective organ system should be disregarded in the calculation of total SOFA and Sepsis. Table 2 shows that the addition of preconditions to the premises caused changes in 40-45% of the cases, however, the derivation correctness of total SOFA scores was 100% without losing performance on examples without preconditions. This shows that fine-tuning can indeed handle exceptions to the rule, opening the doors to possible extensions of our work to include human feedback in the loop.

Table 4: Value correctness of predicted SOFA and SEPSIS scores compared to derivations from real-world clinical measurements. All fine-tuning improvements over one-shot learning are significant.

| | variable | one-shot | one-shot-70B | me-llama | deepseek | fine-tuned | pipeline | multimodal |
|---|---|---|---|---|---|---|---|---|
| current | $S_{CNS}$ | 0.633 | 0.794 | 0.478 | 0.239 | 0.998 | 0.997 | 0.997 |
| | $S_{cardio}$ | 0.405 | 0.6 | 0.249 | 0.273 | 0.985 | 0.971 | 0.987 |
| | $S_{resp}$ | 0.223 | 0.411 | 0.172 | 0.128 | 0.996 | 0.994 | 0.996 |
| | $S_{coag}$ | 0.614 | 0.761 | 0.497 | 0.428 | 0.999 | 0.999 | 0.999 |
| | $S_{liver}$ | 0.492 | 0.809 | 0.502 | 0.208 | 1.000 | 0.999 | 1.000 |
| | $S_{renal}$ | 0.540 | 0.735 | 0.412 | 0.303 | 0.997 | 0.992 | 0.996 |
| | $SOFA_{1:24}$ | 0.111 | 0.236 | 0.076 | 0.044 | 0.979 | 0.958 | 0.981 |
| future | $S_{CNS}$ | 0.532 | 0.625 | 0.408 | 0.231 | 0.701 | 0.732 | 0.760 |
| | $S_{cardio}$ | 0.426 | 0.527 | 0.295 | 0.195 | 0.700 | 0.738 | 0.745 |
| | $S_{resp}$ | 0.201 | 0.348 | 0.163 | 0.082 | 0.746 | 0.754 | 0.780 |
| | $S_{coag}$ | 0.618 | 0.693 | 0.483 | 0.278 | 0.839 | 0.823 | 0.834 |
| | $S_{liver}$ | 0.492 | 0.809 | 0.502 | 0.208 | 0.945 | 0.945 | 0.945 |
| | $S_{renal}$ | 0.482 | 0.620 | 0.365 | 0.211 | 0.553 | 0.451 | 0.473 |
| | $SOFA_{25:48}$ | 0.117 | 0.176 | 0.088 | 0.059 | 0.203 | 0.220 | 0.243 |
| | $SOFA_{diff}$ | 0.139 | 0.167 | 0.128 | 0.127 | 0.151 | 0.147 | 0.163 |
| | SEPSIS Accuracy | 0.834 | 0.857 | 0.763 | 0.692 | 0.868 | 0.873 | 0.886 |
| | SEPSIS Specificity | 0.876 | 0.915 | 0.788 | 0.739 | 0.936 | 0.920 | 0.922 |
| | SEPSIS Sensitivity | 0.3318 | 0.118 | 0.455 | 0.091 | 0.263 | 0.336 | 0.386 |
| | SEPSIS F1 | 0.231 | 0.108 | 0.220 | 0.042 | 0.254 | 0.272 | 0.309 |

**The Bottleneck Lies in Inductive Inference, not Out-of-Distribution Generalization**   Our final evaluation compares the results of partial inference steps to real-world clinical measurements and to SOFA and Sepsis scores derived from these. As shown in the first half of Table 3, the numerical values that the LLM decoder has to extract from time series encodings are nearly perfect for fine-tuning approaches. The second half of Table 3 reveals the bottleneck of teaching medical inference rules to LLMs: Correctness of predicting values of 131 clinical variables 24 hours into the future drops from the high nineties to values in the tens and twenties, especially for sparsely and irregularly observed lab values. The same is true for value correctness of predicted SOFA scores and Sepsis labels: As shown in Table 4, the numerical values produced by mapping clinical measurements in the first 24 hours to SOFA scores are nearly perfect, however, mapping values of forecasted clinical variables to step functions severely impacts correctness, especially on metrics like F1 that are sensitive to imbalanced data. One-shot learning is impacted even more, while fine-tuning results can be improved by moving from a text-based encoding to coupling dedicated TSF encoders with the LLM.

**Discussion**   Leveraging consensus rules that exist for many medical areas, consistent diagnoses for unseen patients and faithful medical inference even including exceptions to the rules can be achieved by fine-tuning LLMs on instantiations of the respective consensus rule system to patient data. In contrast, training or prompting with abstract rules or on very large general-domain data is insufficient. Since an adaptation of an LLM to consensus rules for a specific medical area yields consistent results and faithful medical inference in that area, the bottleneck in consensus-based medical prediction lies in the orthogonal problem of generalization into the future, i.e., in the inherent complexity of forecasting of sparsely and irregularly sampled clinical variables for early prediction. This problem hinders medical prognosis, and needs to be addressed by improved TSF.

## 6 CONCLUSION

LLMs have been shown to excel at various health-related tasks, however, diagnostic AI has not realized the potential to actually reduce the cognitive demand and associated diagnostic errors of clinicians, due to a focus on predictive accuracy of final diagnostic labels (Adler-Milstein et al., 2021). Furthermore, instead of defining diagnostic excellence by perfect labeling based on fully established clinical criteria, Angus & Bindman (2022) advocate to learn patterns of clinical data that are yet unrecognized but still predictive of sepsis. Our approach can be seen as a first step towards these goals by putting a focus on teaching LLMs inference rules that support clinicians, and to learn exceptions to the rules that can capture new clinical patterns. Our experiments show that LLMs perfectly generalize deductive inference to unseen patients, while the bottleneck lies in inductive inference. Future work

shall address the TSF bottleneck of early prediction models, and elevate our work from simulating exceptions to the rule to learning them from real-world clinical data. Furthermore, we will investigate the potential of task association learning (Cai et al., 2025) applied to related consensus definitions.

## REPRODUCIBILITY STATEMENT

This work is based on publicly available data and open-source code. Code to reproduce our experiments is available as supplementary material and will be made public upon acceptance of the paper.

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

# A APPENDIX

## A.1 SOFA SCORE

Table 5: SOFA score for six organ systems, calculated by thresholding corresponding clinical measurements (Vincent et al., 1996). In our setting, we had to recalculate MAP (mean arterial pressure) from SBP and DBP (systolic and diastolic blood pressure), the Horowitz coefficient from PaO2 and FiO2, and had no knowledge about the kind of mechanical ventilation. If no value for calculation in a SOFA subsystem was available, we took a value of 0. Abbreviations: CNS = Central nervous system; GCS = Glasgow Coma Scale; MV = mechanically ventilated including CPAP; MAP = mean arterial pressure, UO = Urine output.

| Score | CNS | Cardiovascular | Respiratory | Coagulation | Liver | Renal |
|---|---|---|---|---|---|---|
| | GCS | MAP or vasopressors | PaO2/FiO2 (mmHg) | Platelets ($\times 10^3/\mu l$) | Bilirubin (mg/dl) | Creatinine (mg/dl) or UO |
| +0 | 15 | MAP $\geq$ 70 mmHg | $\geq$ 400 | $\geq$ 150 | < 1.2 | < 1.2 |
| +1 | 13-14 | MAP < 70 mmHg | < 400 | < 150 | 1.2-1.9 | 1.2-1.9 |
| +2 | 10-12 | dopamine $\leq$ 5 $\mu$g/kg/min OR dobutamine (any dose) | < 300 | < 100 | 2.0-5.9 | 2.0-3.4 |
| +3 | 6-9 | dopamine > 5 $\mu$g/kg/min OR epinephrine $\leq$ 0.1 $\mu$g/kg/min OR norepinephrine $\leq$ 0.1 $\mu$g/kg/min | < 200 AND MV | < 50 | 6.0-11.9 | 3.5-4.9 OR < 500 ml/day |
| +4 | < 6 | dopamine > 15 $\mu$g/kg/min OR epinephrine > 0.1 $\mu$g/kg/min OR norepinephrine > 0.1 $\mu$g/kg/min | < 100 AND MV | < 20 | > 12.0 | > 5.0 OR < 200 ml/day |

## A.2 TIME SERIES FORECASTING

Our TSF model uses the implementation of Staniek et al. (2024) which is based on a Transformer encoder-decoder architecture (Vaswani et al., 2017). First, sparse multivariate input time series are represented as quadruplets $S = \{(f_i, t_i, v_i, n_i)\}_{i=1}^{n}$, where $f_i \in F$ is a clinical variable identifier, $t_i \in \mathbb{R}_{\geq 0}$ is a time index, $v_i \in \mathbb{R}$ the observed value of $f_i$ at $t_i$, and $n_i$ the unique stay identifier. Then the quadruplets for a 24 hour time series are encoded into a dense representation $x$ where every timestep is a vector of feature values representing one hour. We construct this vector by choosing the first observed value during the represented hour for each feature. If no value was observed, we impute zero which corresponds to the mean value due to standardization of the data. Additionally, a mask indicating whether a value was imputed is generated and appended to the vector. For TSF, we use a Transformer model with an autoregressive iterative multistep (IMS) decoder that generates an output vector $\hat{y}_t \in \mathbb{R}^{|F|}$. The predicted output $\hat{y}_t$ is a function of the history $\hat{y}_{<t}$ of predicted timesteps until time $t$, the encoded input $x$, and the model parameters $\theta$: $\hat{y}_t = f_\theta(\hat{y}_{<t}, x)$. To perform long-term TSF using the autoregressive setup, the outputs $\hat{y}_t$ from each time step $t = 1, \ldots, T$ are concatenated. The complete model is trained with masked mean squared error (MSE).

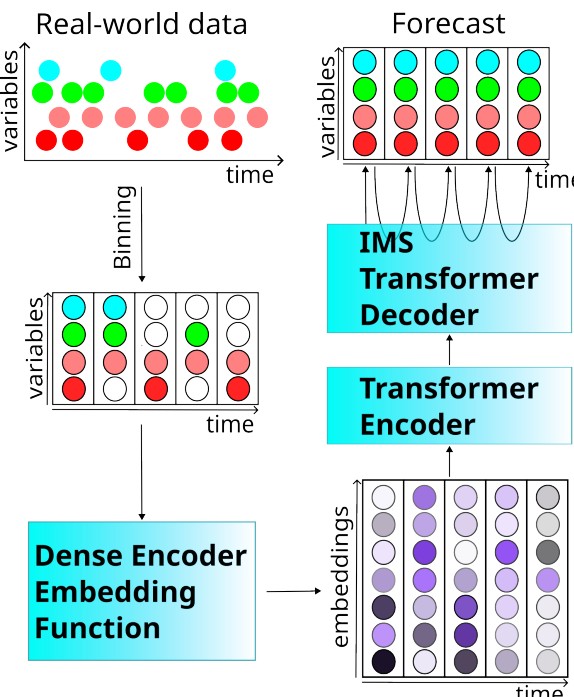

Figure 4: Time Series Forecasting using a dense encoder and iterative multistep decoder architecture.

## A.3 HYPERPARAMETERS AND COMPUTE INFRASTRUCTURE

Table 6: Hyperparameter settings of the time series forecaster.

| Hyperparameter | value |
| --- | --- |
| Embedding Size | 512 |
| Hidden size encoder | 512 |
| Hidden size decoder | 512 |
| # Encoder layers | 2 |
| # Decoder layers | 1 |
| learning rate | 0.0005 |
| attention heads encoder | 8 |
| attention heads decoder | 1 |
| dropout | 0.05 |
| max epochs | 100 |
| patience (early stopping) | 6 |
| Random Seed | unixtime variation |
| # GPUs | 1 |
| Training all clock time | 5 hours |
| GPU type | Nvidia GTX 1080 Ti |

Table 7: Hyperparameter settings of the multi-modal and LLM finetuning architectures.

| Hyperparameter | value |
| --- | --- |
| optimizer | adam |
| learning rate | 0.000002 |
| max epochs | 10 |
| connector first layer | 131x2096 |
| connector second layer | 2096x4096 |
| lora-r | 16 |
| lora-alpha | 16 |
| lora-dropout | 0.1 |
| lora-target-modules | all-linear |
| # GPUs | 1 |
| Training all clock time | 40 hours |
| GPU type | NVIDIA A100 |
| Starting model | meta-llama/Llama-3.1-8B-Instruct |

## A.4 CLINICAL FEATURES

Table 8: Feature list for MIMIC-III: Besides the following 131 dynamic variables, only age and gender were extracted. The 15 variables marked with an asterisk are directly used for calculating the SOFA score.

| | | | |
|---|---|---|---|
| ALP | Epinephrine* | LDH | Packed RBC |
| ALT | Famotidine | Lactate | Pantoprazole |
| AST | Fentanyl | Lactated Ringers | Phosphate |
| Albumin | FiO2* | Levofloxacin | Piggyback |
| Albumin 25% | Fiber | Lorazepam | Piperacillin |
| Albumin 5% | Free Water | Lymphocytes | Platelet Count* |
| Amiodarone | Fresh Frozen Plasma | Lymphocytes (Absolute) | Potassium |
| Anion Gap | Furosemide | MBP | Pre-admission Intake |
| BUN | GCS_eye* | MCH | Pre-admission Output |
| Base Excess | GCS_motor* | MCHC | Propofol |
| Basophils | GCS_verbal* | MCV | RBC |
| Bicarbonate | GT Flush | Magnesium | RDW |
| Bilirubin (Direct) | Gastric | Magnesium Sulfate (Bolus) | RR |
| Bilirubin (Indirect) | Gastric Meds | Magnesium Sulphate | Residual |
| Bilirubin (Total)* | Glucose (Blood) | Mechanically ventilated | SBP* |
| CRR | Glucose (Serum) | Metoprolol | SG Urine |
| Calcium Free | Glucose (Whole Blood) | Midazolam | Sodium |
| Calcium Gluconate | HR | Milrinone | Solution |
| Calcium Total | Half Normal Saline | Monocytes | Sterile Water |
| Cefazolin | Hct | Morphine Sulfate | Stool |
| Chest Tube | Heparin | Neosynephrine | TPN |
| Chloride | Hgb | Neutrophils | Temperature |
| Colloid | Hydralazine | Nitroglycerine | Total CO2 |
| Creatinine Blood* | Hydromorphone | Nitroprusside | Ultrafiltrate |
| Creatinine Urine | INR | Norepinephrine* | Urine* |
| D5W | Insulin Humalog | Normal Saline | Vancomycin |
| DBP* | Insulin NPH | O2 Saturation | Vasopressin |
| Dextrose Other | Insulin Regular | OR/PACU Crystalloid | WBC |
| Dobutamine* | Insulin largine | PCO2 | Weight |
| Dopamine* | Intubated | PO intake | pH Blood |
| EBL | Jackson-Pratt | PaO2* | pH Urine |
| Emesis | KCl | PT | |
| Eoisinophils | KCl (Bolus) | PTT | |

## A.5 ONE-SHOT PROMPT

For the given 24 hours of measurements, apply the definitions of the SOFA score given by the following table:

```
{| class="wikitable"
!
!Central nervous system
!Cardiovascular system
!Respiratory system
!Coagulation
!Liver
!Renal function
|-
!Score
! [[Glasgow coma scale]]
! Mean arterial pressure OR administration of vasopressors required
! PaO2/FiO2 <nowiki>[mmHg (kPa)]</nowiki>
! Platelets (x103/ul)
! Bilirubin (mg/dl) [umol/L]
! Creatinine (mg/dl) [umol/L] (or urine output)
|-
! +0
| 15 || [[Mean_arterial_pressure|MAP]] >= 70 mmHg || >= 400 (53.3) || >= 150 || < 1.2 [< 20] || < 1.2 [<
↪   110]
|-
! +1
| 13-14 || MAP < 70 mmHg || < 400 (53.3) || < 150 || 1.2-1.9 [20-32] || 1.2-1.9 [110-170]
|-
! +2
| 10-12 || [[dopamine]]  <= 5 ug/kg/min or [[dobutamine]] (any dose) || < 300 (40) || < 100 || 2.0-5.9
↪   [33-101] || 2.0-3.4 [171-299]
|-
! +3
| 6-9 || dopamine > 5 ug/kg/min OR [[epinephrine]] <= 0.1 ug/kg/min OR [[norepinephrine]]  <= 0.1 ug/kg/min
↪   || < 200 (26.7) '''and''' mechanically ventilated including CPAP || < 50 || 6.0-11.9 [102-204] ||
↪   3.5-4.9 [300-440] (or < 500 ml/day)
|-
! +4
| < 6 || dopamine > 15 ug/kg/min OR epinephrine > 0.1 ug/kg/min OR norepinephrine > 0.1 ug/kg/min || < 100
↪   (13.3) '''and''' mechanically ventilated including CPAP || < 20 || > 12.0 [> 204] || > 5.0 [> 440] (or <
↪   200 ml/day)
|}
```

A patient is septic if suspected infection is positive and the future total SOFA score calculated with a best guess on how the values develop is two points (or more) higher than the current SOFA score.
Only answer like in the given example. Here is the example:
Patient is 63.0 years old and is male. Given all the information in this text, answer the question at the end.
Here are the measurements: DBP at time -23.68: 36.0, SBP at time -23.68: 71.0...
Now answer the following question in the given format:
Doctors suspect an infection, based on this information and the other information in this text, will the patient be classified as septic tomorrow?
First we need to calculate the SOFA scores given the extracted values. The SOFA scores for the current time are the following:
The minimum value of GCS_eye is 4.0, GCS_motor is 6.0 and GCS_verbal is 5.0, this produces the sum 15.0 and means the CNS SOFA is 0.
Because minimum MAP is 43.333, max Dopamine is 0, max Dobutamine is 0, max Epinephrine is 0 and max Norepinephrine is 0 with a patient weight of 80 kg, the cardiovascular SOFA is 1.
Given that minimum PO2 is 141.0 and minimum FiO2 is 1 the calculated PAO2FIO2 is 141.0, this means the respiratory SOFA is 3.
Because the minimum Platelet count is 235.0 the coagulation SOFA is 0.
The maximum Bilirubin (Total) is 1.8 leading to a liver SOFA of 1.
Because total Urine output is 1585.0 and maximum creatinine in the blood is 1.4 the renal SOFA is 1.
To summarize: the patient has a total SOFA score of 6.
Now we need to calculate the SOFA scores with forecasted values. The SOFA scores in the future based on the forecasted values are the following:
The minimum value of GCS_eye will be 4.0, GCS_motor will be 6.0 and GCS_verbal will be 5.0, this produces the sum 15.0 and means the CNS SOFA will be 0.
Because future minimum MAP will be 55.667, future max Dopamine will be 0, future max Dobutamine will be 0, future max Epinephrine will be 0 and future max Norepinephrine will be 0 with a patient weight of 80 kg, the cardiovascular SOFA will be 1.
Given that minimum PO2 will be 141.0 and minimum FiO2 will be 1 the forecasted PAO2FIO2 will be 141.0, this means the respiratory SOFA will be 3.
Because the Platelet count will be 295.0 the coagulation SOFA is going to be 0. The maximum Bilirubin (Total) will be 1.8 leading to a liver SOFA of 1.
Because Urine output will be 1635.0 and maximum creatinine in the blood will be 1.1 the renal SOFA will be 0.
To summarize: the patient will have a future total SOFA score of 5.
The patient will not develop sepsis in the next 24 hours, because total SOFA increased only by -1 and infection is suspected.
The example is now finished. Say "The patient will develop sepsis" in the last sentence if the criteria are met (if total SOFA changed by 2 and infection is suspected).
Patient is 76.0 years old and is female. Given all the information in this text, answer the question at the end.
Here are the measurements: DBP at time -22.97: 56.0, GCS_eye at time -22.97: 4.0, GCS_motor at time -22.97: 6.0...
Now answer the following question in the given format:
The doctors don't suspect an infection, based on this information and the other information in this text, will the patient be classified as septic tomorrow?

## A.6 SETUP FOR LEARNING FROM PRECONDITIONS

Table 9: Medical preconditions for five organ systems indicated by ICD-10 codes. In-distribution (ID) data were seen, out-of-distribution (OOD) data were unseen during fine-tuning.

|  | lung | kidney | coagulation | liver | cardiovascular |
|---|---|---|---|---|---|
| ID | J40, J41, J42 | N18.9, N28 | D68.4, D68.5 | K70.0, K70.41 | I50.0, I50.9 |
| OOD | J44.9 | N19 | D68.6 | K70.3 | I50.1 |

Table 10: ICD-10 Code frequency for in- and out-of-distribution testsets.

| ICD-10 Code | # ID Test | # OOD Test |
|---|---|---|
| J40 | 263 | 194 |
| J41 | 239 | 173 |
| J42 | 253 | 194 |
| J44.9 | - | 166 |
| N18.9 | 266 | 172 |
| N19 | - | 204 |
| N28 | 281 | 151 |
| D68.4 | 242 | 171 |
| D68.5 | 244 | 162 |
| D68.6 | - | 176 |
| I50.0 | 264 | 185 |
| I50.1 | - | 193 |
| I50.9 | 229 | 162 |
| K70.0 | 230 | 165 |
| K70.3 | - | 162 |
| K70.41 | 215 | 182 |
| No-Preexisting | 274 | 189 |

## A.7 DISCUSSION OF 5% ERROR MARGIN

The 5% margin for deviations of predictions is justified by the statistics from our experimental results. For example, $SOFA_{25:48}$ score ranges between 0 and 24. The experimental results in Figure 5 plot scores obtained by dividing predicted values by gold standard values for this metric, showing that a 10% margin (dotted vertical lines) would include false positives, while a 5% margin does not. Similar results are obtained for other predictions.

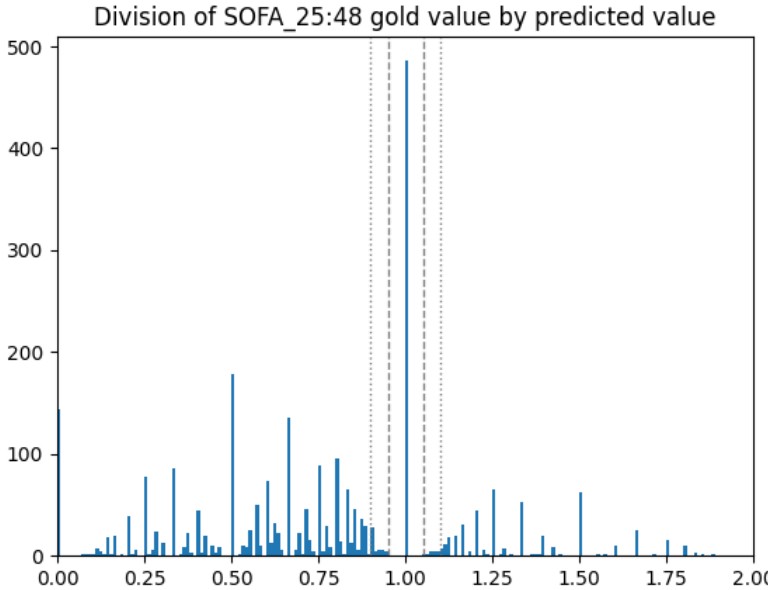

Figure 5: Histogram of the scores obtained by dividing predicted by gold standard values for $SOFA_{25:48}$ score. Dashed vertical lines show the 5% interval, dotted vertical lines show the 10% interval

