# OpenReview forum: "Teaching Consensus Rules (and Exceptions) to LLMs for Trustworthy Medical Reasoning"
_ICLR.cc/2026/Conference — Submitted to ICLR 2026_

### Official Review · Reviewer_g5bc · 2025-10-14

**Soundness:** 3
**Presentation:** 2
**Contribution:** 2
**Rating:** 2
**Confidence:** 4

**Summary:**

The paper proposes teaching LLMs to follow medical consensus rules (deductive steps) and handle exceptions, using Sepsis-3 as a worked example. The authors fine-tune an 8B Llama-3 model on verbalized, step-by-step rule instantiations paired with patient data, add synthetic exceptions via ICD-10 codes, and couple the LLM with a time-series forecasting (TSF) model (pipeline or multimodal) to predict 24h-ahead clinical variables. They evaluate derivation correctness, forced derivation correctness, and value correctness, reporting near-perfect derivation correctness after fine-tuning, but much weaker value correctness for future variables/SOFA and modest Sepsis F1.

**Strengths:**

Fine-tuned 8B consistently beats one-shot (8B/70B) and a medical-pretrained baseline on derivation correctness, including forced-history tests.
Thoughtful framing of the TSF bottleneck and exploration of pipeline vs multimodal coupling with a forecaster.

**Weaknesses:**

1.Clinical utility is limited by forecasting. Future SOFA value correctness is low; Sepsis F1 tops at ~0.31 with low sensitivity, which is problematic for early-warning use. Stronger TSF and equal-compute comparisons are needed.
2.Synthetic exceptions only. The “exception” mechanism is shown on synthetic ICD scenarios; no real clinician-in-the-loop edits or real exception logs are used, so robustness and safety are uncertain.
3.Parser-coupled evaluation. The metrics depend on output formatting and a numeric tolerance; this risks over-estimating competence if the model overfits the template. Ablations on phrasing/format would help.
4.Potential leakage and narrow scope. “Suspected infection” is provided as input while also defining the label; even if aligned with prior labeling schemes, a leakage-control analysis is warranted. The study is single-center, single-definition (Sepsis-3) and doesn’t test other consensus families.

**Questions:**

1.Leakage control: Since “suspected infection” is both input and label component, can you report experiments excluding it from inputs, or using delayed/uncertain infection indicators?
2.Evaluation robustness: How sensitive are derivation/value correctness scores to prompt paraphrases and output formatting? Please include format-randomized prompts and parser-free checks (e.g., structured outputs).
3.Real exceptions: Can you provide a small real-world clinician-edited set (or retrospective EHR notes) to validate the exception mechanism beyond synthetic ICD insertions?

---

> ### Author Response · Authors · 2025-11-20
> **# Reply to Reviewer g5bc**
>
> Thank you for your insightful review. Please find answers to your questions below.
>
>  1. **Leakage control:** Since the purpose of our work is show that the standard usage of LLMs in healthcare -- prompting pre-trained LLMs to perform medical question-answering -- is insufficient when it comes to early prediction tasks, we focused on the deductive and inductive inference steps that are necessary for early prediction, while simplifying the task by fixing the Sepsis onset to a given onset of suspicion of infection. This is consistent with other widely cited work (see Nemati et al. 2018) and does not introduce leakage since the final decision of Sepsis yes/no relies on the computation of SOFA scores and their difference.
>
>  2. **Evaluation robustness against prompt variation:** We updated all one-shot uses of LLMs with a more sophisticated prompt template that includes an explicit verbalization of the abstract consensus definition. The results for the prompted models improved only slightly, confirming our main claim of the necessity of fine-tuning for accurate and interpretable early prediction in healthcare. The new one-shot prompt is shown in Appendix A5 in the updated paper.
>
>
> 3. **Real exceptions:** There is clear evidence that incorporating pre-existing medical conditions (comorbidities / chronic organ failure) can improve sepsis prediction (Sarraf et al. 2024 https://doi.org/10.1016/j.jcrc.2024.154857; Christensen et al. 2023 https://doi.org/10.1097/CCE.0000000000000865), however, chronic organ disfunction is still a complicated issue and was not considered in the recent update of SOFA-2 (Ranzani et al. 2025 https://doi.org/10.1001/jama.2025.20516). Our inclusion of pre-existing conditions is thus most conservative in that it recognizes an exception due to medical preconditions, however, since there is no consensus on consequences of organ disfunctions, a safe exception is to disregard the SOFA score for the respective organ system. However, learning real exceptions from clinician-edited data is an important point for future work.
>
> Finally, we would like to stress that a focus on Sepsis-3 is not necessarily a limiting factor of our work: The Sepsis-3 consensus definition is a composition of several other consensus definitions: It subsumes the SOFA definition which itself can be seen as a combination of consensus definitions for each of 6 organ systems (for example, the NDIGO definition of acute kidney injury (AKI) is similar to the definition of kidney SOFA, etc.) and adds complexity by computing a change of at least two points in 24 hours. From a mathematical point of view, most consensus definitions map thresholds on input measurements to step functions, thus our investigation of Sepsis-3 covers a wide variety of consensus definitions, and adds a prognostic aspect that is implicit in most other definitions.

---

### Official Review · Reviewer_3fBQ · 2025-10-29

**Soundness:** 2
**Presentation:** 2
**Contribution:** 2
**Rating:** 2
**Confidence:** 2

**Summary:**

The methodology transforms medical consensus guidelines (e.g., Sepsis-3/SOFA) into auto-generated "reasoning chains (scratchpads)" for LLM fine-tuning, enabling step-by-step rule-based calculations (e.g., extrema extraction, threshold mapping). Two metrics are introduced: derivation correctness (faithful rule-based inference) and value correctness (alignment with real-world data within error margins). An exception-handling mechanism uses synthetic ICD-10 histories to test "rules + exceptions" learnability, including OOD scenarios. The approach employs Llama-3 8B with LoRA fine-tuning and a Transformer-based TSF model, integrated via pipeline (prompt appending) or multimodal (embedding fusion). Results show near-perfect derivation correctness post fine-tuning, outperforming one-shot models (including 70B) and pre-trained medical LLMs. Exception learning achieves 100% correctness in ID/OOD, while value correctness for current variables is strong but declines for future predictions, revealing TSF as a bottleneck. Innovations include compositional rule learning, automated reasoning faithfulness evaluation, a trainable "rules + exceptions" framework, and TSF-LLM integration to enhance forecasting.

**Strengths:**

- This method shows potential as a foundational module for "guideline adherence + human-AI collaboration." The study highlights the bottleneck in TSF, pointing out clear directions for future improvements.
- Task definitions, rule diagrams (Figure 3), and examples (Figure 2) are intuitive; appendices provide clear SOFA tables, feature lists, hyperparameters, and hardware details.

**Weaknesses:**

- Inconsistent handling of missing TSF values: The main text (Sec.4) states that "missing values are only forward-filled from the previous day (carry forward)," while Appendix A.2 describes "hourly vectorization, where missing values are set to 0 (standardized mean) with an added missing mask."
- Forced derivation correctness interpretation: The paper claims "correct histories and predicted histories are very similar," which holds for fine-tuned models but not for one-shot models (Table 1 shows notable differences between forced and non-forced one-shot results, with SOFAdiff performing worse under forced conditions). A more precise explanation and analysis of the reasons are needed.
- Pipeline-related drop in current urine output: In Table 3, the pipeline approach shows a significant drop in current urine output performance (0.966 → 0.761), which remains unexplained.

**Questions:**

Which policy is actually used—LOCF (last observation carried forward) or mean imputation with a missingness mask? Are training and inference consistent? If both were tried, please provide an ablation and discuss the impact on results.

For training targets of future variables/future SOFA, do you use the ground-truth observations from the second 24h window, or fixed TSF predictions? Please clarify to rule out any information leakage and specify whether teacher forcing or scheduled sampling is used.

How exactly do you “force up to the last measurement token” (e.g., constrained decoding, special delimiters, alignment with a parser)? Why do one-shot models show lower SOFAdiff/SEPSIS performance under the forced setting than the non-forced setting? Is this due to alignment/parsing artifacts, truncation, or differences in node matching?

---

> ### Author Response · Authors · 2025-11-20
> **# Reply to Reviewer 3fBQ**
>
> Thank you for your insightful review. Please find answers to your questions below.
>
> **Imputation of missing values:** The statement in Section 4 that "missing values are being carried forward from the previous day only" refers to the deterministic computation of the ground truth from real-world clinical measurements. For prediction purposes, either no imputation (LLM) or mean imputation in the connected TSF model (LLM + TSF) is used. We apologize for the unclear description which is fixed in the updated paper.
>
> **Teacher Forcing versus Student Forcing:** We use the model's own previous predictions as context for next time step prediction at training and inference time. This "student forcing" strategy has shown to outperform teacher forcing as well as scheduled sampling in experiments with the TSF model that we connect to our LLM.
>
> **Forced Decoding:** We use a constrained decoding strategy to compute "forced derivation correctness" during evaluation. This is done by using the ground-truth history during decoding (similar to a teacher forcing strategy used in training).
>
> To understand the perceived drop of performance in SOFA_diff and SEPSIS, imagine a model that produces for every patient values of current SOFA=2 and future_SOFA=4. The SOFA_diff will always yield a value of 2, yielding correct derivation correctness for SOFA_diff in all cases, even if the predictions for current and future SOFA are wrong.
> However, if the model is forced to use the gold standard value, for example, future_SOFA=8 and current_SOFA=4, outputting SOFA_diff=2 does not yield a correct prediction anymore. This results in a lower forced derivation correctness. Forced derivation correctness is particularly impactful for SOFA_diff and SEPSIS because the difference or binary classification are not as varied as the other variables.

---

### Official Review · Reviewer_hHAT · 2025-10-29

**Soundness:** 2
**Presentation:** 4
**Contribution:** 2
**Rating:** 2
**Confidence:** 4

**Summary:**

The paper investigates how machine learning for early prediction in medicine can be made more explainable and faithful. The authors aim to teach LLMs to follow consensus rules in their reasoning, these are rules that specify how practitioners proceed to diagnose certain diseases. To achieve this they finetune existing small models on this task and compare it against existing larger LLMs augmented with "one-shot" learning as well as models trained on medical text including consensus definitions. They limit their analysis and evaluation to the Sepsis-3 consensus definition. They find that small fine-tuned models perform almost perfect outperforming bigger models. To improve the forcasting using  sparsely and irregularly sampled clinical variables, they propose to use representations of a time series
forecasting model attached to the LLM using a trained conntector i.e. in a multimodal setting.

**Strengths:**

- The paper is clearly written and easy to follow
- The problem statement is clearly motivated
- The promised claims are supported

**Weaknesses:**

-  The paper is primarily an application of fine-tuning for reasoning and using a connector to embed a time-series forecasting (TSF) model. Hence, the contribution is limited. Although nobody has looked at this particular problem, it is rather unsurprising that this works well.

- The paper only focuses on one specific application, namely Sepsis-3.

- LLMs are known to be sensitive to the system prompt, and according to the experimental setup there was no prompt engineering performed. The “one-shot” prompt is shown in Appendix A.5, which suggests that the prompt lacks a sophisticated structure. This significantly weakens the claims, as performance could likely be improved by using a more sophisticated prompt template or more than one ICL example.

- The dataset is very commonly used; hence, it would not be surprising if there were leaks into the training set. Because the Llama 3 dataset is not fully public, focusing on models such as OLMo would have been better.

- The analysis misses a nuanced evaluation of failure modes—for example, whether the outputs of the untuned model are near-misses or whether the reasoning is completely off. This is very important to understand what fine-tuning is doing.

- Given that the goal is to investigate “reasoning,” it is not clear why you did not compare against open-source reasoning models such as Qwen.

- Moreover, it would be important to compare to methods such as DSPy or TextGrad, as these methods have been shown to be efficient while not requiring changes to model weights.

Minor:
- The tables are hard to parse. Please bold the best values and also round the results.

**Questions:**

None.

---

> ### Author Response · Authors · 2025-11-20
> **# Reply to Reviewer hHAT**
>
> Thank you for your insightful review. Please find replies to your comments below.
>
> **Contribution of paper:** The contribution of our paper is to show that the standard usage of LLMs in healthcare -- prompting pre-trained LLMs to perform medical question-answering -- is insufficient when it comes to early prediction tasks. Here fine-tuning significantly outperforms prompted LLMs that are an order or magnitude larger, even if the prompt includes the explicit consensus definition and an instantiation to patient data. Fine-tuning also can learn exceptions to the consensus rules. Furthermore, we point out that the remaining problem in early prediction is not out-of-distribution generalization, but the orthogonal problem of generalization into the future. This problem is addressed by connecting a TSF model to LLMs.
>
> **Consensus definitions beyond Sepsis-3:** We focused on the Sepsis-3 consensus definition since it is a composition of several other consensus definitions: It subsumes the SOFA  definition which itself can be seen as a combination of consensus definitions for each of 6 organ systems (for example, the NDIGO definition of acute kidney injury (AKI) is similar to the definition of kidney SOFA, etc.) and adds complexity by computing a change of at least two points in 24 hours. From a mathematical point of view, most consensus definitions map thresholds on input measurements to step functions, thus our investigation of Sepsis-3 covers a wide variety of consensus definitions, and adds a prognostic aspect that is implicit in most other definitions.
>
> **More sophisticated prompt:** We updated all one-shot uses of LLMs with a more sophisticated prompt template that includes an explicit verbalization of the abstract consensus definition. The results for the prompted models improved only slightly, confirming our main claim of the necessity of fine-tuning for accurate and interpretable early prediction in healthcare. The new one-shot prompt is shown in Appendix A5 in the updated paper.
>
> **Comparison to other LLMs:** In addition to updating the one-shot prompt with explicit consensus rules, we performed one-shot experiments with reasoning LLMs (deepseek https://huggingface.co/deepseek-ai/DeepSeek-R1-Distill-Llama-8B). Please see the additional column in Tables 1, 3, and 4 in the updated paper.
>
>
> **Nuanced evaluation:**
> We added a histogram showing that the 5% slack in our evaluation metrics accurately captures the majority of small errors (see Appendix A7 in the updated paper).

---

> > ### Comment · Reviewer_hHAT · 2025-11-23
> >
> > Thank you for the elaborate rebuttal. Please find my answers below.
> >
> > **Contribution of the paper**: I can see that this is what you intended to show, yet the claim is actually weaker.
> > You demonstrate that small LLMs struggle with formulating diagnoses, which is an important setting to consider.
> > However, to make a broader claim that current LLMs cannot follow consensus guidelines, larger and potentially closed-source models would need to be evaluated.
> >
> > Yet it is not surprising that fine-tuning works better than simply prompting; this is a well-known and established fact.
> > I would kindly invite you to elaborate on the specific contribution for this exact setting. Are there prior studies indicating that fine-tuning (FT) for healthcare applications can be problematic?
> >
> > **Consensus definitions beyond Sepsis-3**:
> > That is a fair argument. Yet making claims that LLMs cannot follow medical consensus rules in general requires a broader evaluation than just one setting.
> >
> > **More sophisticated prompt:**
> > I can see that you changed the system prompt. However, it still lacks the fine-grained instructions and conditioning that strong system prompts have. I would kindly suggest checking out the system prompts of common AI tools, e.g., https://github.com/x1xhlol/system-prompts-and-models-of-ai-tools.
> >
> > **Comparison to other LLMs:**
> > Thanks for including the extra evaluation.
> >
> > **Nuanced evaluation:**
> > Thanks for including the extra evaluation metrics. However, they do not show which failure modes are fixed by fine-tuning. To cite my initial review: "whether the outputs of the untuned model are near-misses or whether the reasoning is completely off.", is not clarified by your new evaluation.
> >
> > For the reasons mentioned above, I will keep my score.

---

> ### Author Response · Authors · 2025-11-26
> **Reply to Reviewer hHAT**
>
> **Contribution of the paper** You write that "fine-tuning works better than simply prompting; this is a well-known and established fact." Looking at recent surveys of the field of medical reasoning (https://arxiv.org/abs/2508.00669), we respectfully disagree. We did not find any concurrent work that uses medical consensus definitions to automatically create reasoning traces for fine-tuning (in contrast to fine-tuning on manually created medical diagnoses, which is not scalable) and presents a thorough automatic evaluation of the inductive and deductive aspects of medical inference against gold standard reasoning traces (instead of only evaluating on the final answer, which still is the standard, or conducting manual evaluations).
>
> We think that what counts here is the idea! Broadening the evaluation to other consensus definitions or using open-source models larger than 70B is impossible during the rebuttal period, and comparing to larger closed-source commercial LLMs is prohibited by the data privacy policy of PhysioNet.

---

### Official Review · Reviewer_fNHn · 2025-11-01

**Soundness:** 3
**Presentation:** 3
**Contribution:** 3
**Rating:** 4
**Confidence:** 3

**Summary:**

The paper aims to train language models to follow medical consensus guidelines step-by-step when reasoning about diagnoses and predictions, focusing on transparent and faithful inference rather than only accuracy. It uses verbalized consensus rules (explicit textual versions of medical inference steps) linked to patient records to fine-tune large language models so that they learn both standard rules and their exceptions. The authors demonstrate their approach using the Sepsis-3 definition, which combines deductive rules (if-then mappings for organ dysfunction) with inductive ones (time-series forecasting of clinical variables). They generate textual training examples where each inference step (such as computing SOFA subscores or combining them into a sepsis label) is verbalized and paired with patient data. Experiments on MIMIC-III show that small fine-tuned models achieve nearly perfect derivation correctness and outperform much larger one-shot or pretrained medical LLMs under all evaluation metrics. They also introduce evaluation metrics distinguishing derivation correctness (logical faithfulness of reasoning) from value correctness (numerical accuracy of outputs). The paper finds that the key challenge for early prediction is not domain generalization but forecasting future, irregularly sampled measurements, which can be mitigated by combining LLMs with a time-series forecaster in a multimodal setup.

**Strengths:**

Below are the strengths of the paper in my opinion:

1. Clear and reproducible fine-tuning framework where LLMs are trained on verbalized consensus rules instantiated to patient data.
2. The distinction between derivation correctness and value correctness provides a structured and transparent way to evaluate the model’s inference faithfulness and numerical accuracy.
3. The integration of a time-series forecasting model with the LLM in a multimodal setup directly addresses the challenge of predicting future, irregularly sampled clinical variables that is identified as the core issue in the paper.
4. The experimental setup uses a publicly available dataset (MIMIC-III) and applies deterministic consensus-based labeling, ensuring internal consistency between training and evaluation and reproducibility.
5. The inclusion of rule exceptions (via ICD-10 preconditions) introduces a controlled mechanism to test model behavior under rule deviations.

**Weaknesses:**

I find the methodology looks very appropriate in the current realm of medical AI, however, there are some limitations that addressing them is missing in the current iteration of the work. The main weakness is about generalizability and can be addressed by looking beyond Sepsis-3. Below is a list of weaknesses about this work:

1. Although sepsis is very important but as a tool, the evaluation focuses on a single medical consensus guideline (Sepsis-3), limiting evidence that the proposed approach generalizes to other rule systems or disease contexts.
2. The paper lacks ablation studies to quantify the individual contributions of verbalization, fine-tuning, and multimodal integration to the final performance.
3. The generation of verbalized training data depends on manually designed templates and synthetic exception cases, introducing potential bias and limiting scalability.
4. The validation of derived reasoning chains relies on deterministic correctness checks rather than expert or clinical outcome validation, leaving open whether the reasoning aligns with real expert judgment.
5. While derivation and value correctness are well-defined, the methodology does not discuss uncertainty estimation or statistical significance of the reported near-perfect scores. (Bootstrapping can be used here for example).

**Questions:**

My main questions are directly related to the weaknesses I raised above.

1. How well does the proposed method generalize to other medical consensus definitions beyond Sepsis-3, particularly those that rely on qualitative or interview-based assessments rather than numerical thresholds?
2. Could you provide quantitative evidence or ablations to isolate the impact of fine-tuning on verbalized rules versus the multimodal integration with the time-series forecaster?
3. How were exceptions to the consensus rules (e.g., ICD-10 preconditions) validated to ensure that they represent realistic clinical edge cases rather than synthetic artifacts?
4. Could you clarify how sensitive the metrics are to small errors in the reasoning chain and whether they correlate with overall diagnostic accuracy?
5. How much human curation is required to construct these templates, and could the process be automated for other diseases?
6. Did the authors assess how robust the model is to noisy or incomplete clinical measurements, which are common in real EHR data?
7. And lastly for a minor comment: What are the main limitations that prevent the current system from being directly applicable to clinical decision support, and how do the authors envision addressing them?

**Details Of Ethics Concerns:**

n/a.

---

> ### Author Response · Authors · 2025-11-20
> **# Reply to Reviewer fNHn**
>
> Thank you for your insightful review. Please find answers to your questions below.
>
> 1. **Consensus definitions beyond Sepsis-3:** We focused on the Sepsis-3 consensus definition since it is a composition of several other consensus definitions: It subsumes the SOFA definition which itself can be seen as a combination of consensus definitions for each of 6 organ systems (for example, the NDIGO definition of acute kidney injury (AKI) is similar to the definition of kidney SOFA, etc.) and adds complexity by computing a change of at least two points in 24 hours. From a mathematical point of view, most consensus definitions map thresholds on input measurements to step functions, thus our investigation of Sepsis-3 covers a wide variety of consensus definitions, and adds a prognostic aspect that is not explicit in most other definitions.
>
> 2. **Ablation study:** Our Tables 1, 3, and 4 *do* in fact present ablations. The one-shot model (first column) is the baseline model that is fine-tuned without a separate TSF module (fifth column), or TSF modules are combined with the LLM in a pipeline setup (sixth column) or a multimodal setup (seventh column). The ablation shows improvements with each column.
>
>
> 3. **Exceptions to the rule:** There is clear evidence that incorporating pre-existing medical conditions (comorbidities / chronic organ failure) can improve sepsis prediction (Sarraf et al. 2024 https://doi.org/10.1016/j.jcrc.2024.154857; Christensen et al. 2023 https://doi.org/10.1097/CCE.0000000000000865), however, chronic organ disfuntion is still a complicated issue and was not considered in the recent update of SOFA-2 (Ranzani et al. 2025 https://doi.org/10.1001/jama.2025.20516). Our inclusion of pre-existing conditions is thus most conservative in that it recognizes an exception due to medical preconditions, however, since there is no consensus on consequences of organ disfunctions, a safe exception is to disregard the SOFA score for the respective organ system.
>
> 4. **Sensitivity of metrics:** We added a histogram showing that the 5% slack in our evaluation metrics accurately captures the majority of small errors (see Appendix A7 in the updated paper). Furthermore, we would like to stress that statistical significance was already assessed in the submission version (see Tables 3 and 4).
>
> 5. **Templates:** Templates to instantiate verbalized consensus definitions to patient data can easily be constructed using LLMs. In fact, the purpose of our text-based approach is not only direct interpretability by medical practitioners, but also easy transfer of consensus definitions to patient data for the purposes of fine-tuning. We emphasized this fact in the updated version of the paper.
>
> 6. **Real EHR data:** Our input data *are* real EHR data with all the complications of sparse and irregular sampling. The clinical measurements for the first 24 hours are taken as is from the MIMIC dataset. For the second day, missing values are carried forward from the first day.
>
> 7. **Applicability to clinical decision support:**  Integrating our system for early prediction of sepsis into clinical practice is definitely the main future goal of our work. At the moment, the limiting factor is still the accuracy of the prediction (value correctness). Our future research will focus on this aspect.

---

### Meta-Review · Area_Chair_JeWk · 2026-01-09

**Summary:**

While the paper is clearly written and technically sound, the main contribution largely confirms a well-known result, that fine-tuning on structured rules outperforms prompting; the evaluation is narrowly confined to a single consensus definition (Sepsis-3), and shows limited clinical utility in the most relevant setting (future prediction), leaving concerns about generalization, and broader impact unresolved.
The work itself has merit, however the authors should consider submitting the work to a domain-specific conference for example: Machine Learning for Healthcare (MLHC) or even 7th Annual Conference on Health, Inference, and Learning (CHIL 2026).
With  focus on a  single medical consensus guideline (Sepsis-3), it is unclear if the proposed approach generalizes to other rule systems or disease contexts - generalization is key in learning focused conferences, thus the authors may consider the above-suggested venues.

**Reviewer Concerns:**

Ablations and statistical robustness

**Reviewer Scores:**

unchanged

---

### Decision · Program_Chairs · 2026-01-26

Reject